# The Extension of Surgery Predicts Acute Postoperative Pain, While Persistent Postoperative Pain Is Related to the Spinal Pathology in Adolescents Undergoing Posterior Spinal Fusion

**DOI:** 10.3390/children9111729

**Published:** 2022-11-10

**Authors:** Tommi Yrjälä, Ilkka Helenius, Tiia Rissanen, Matti Ahonen, Markku Taittonen, Linda Helenius

**Affiliations:** 1Department of Anesthesia and Intensive Care, University of Turku and Turku University Hospital, 20521 Turku, Finland; 2Department of Orthopedics and Traumatology, University of Helsinki and Helsinki University Hospital, 00029 Helsinki, Finland; 3Department of Biostatistics, University of Turku, 20500 Turku, Finland; 4Department of Pediatric Surgery, Orthopedics and Traumatology, University of Helsinki and Helsinki University Hospital, 00029 Helsinki, Finland

**Keywords:** adolescent idiopathic scoliosis, Scheuermann kyphosis, spondylolisthesis, posterior spinal fusion, postsurgical pain

## Abstract

Persistent pain after posterior spinal fusion affects 12 to 42% of patients with adolescent idiopathic scoliosis. The incidence of persistent pain among surgically treated children with Scheuermann kyphosis and spondylolisthesis is not known. The aim of our study was to determine the predictors and incidence of acute and chronic postoperative pain in adolescents undergoing posterior spinal fusion surgery. The study was a retrospective analysis of a prospectively collected pediatric spine register data. The study included 213 consecutive patients (158 AIS, 19 Scheuermann kyphosis, and 36 spondylolisthesis), aged 10–21 years undergoing posterior spinal fusion at a university hospital between March 2010 and March 2020. The mean (SD) daily postoperative opioid consumption per kilogram was significantly lower in the spondylolisthesis patients 0.36 mg/kg/day (0.17) compared to adolescent idiopathic scoliosis 0.51 mg/kg/day (0.25), and Scheuermann kyphosis 0.52 mg/kg/day (0.25) patients after surgery (*p* = 0.0004). Number of levels fused correlated with the daily opioid consumption (r_s_ = 0.20, *p* = 0.0082). The SRS-24 pain domain scores showed a statistically significant improvement from preoperative levels to two-year follow-up in all three groups (*p* ≤ 0.03 for all comparisons). The spondylolisthesis patients had the lowest SRS pain domain scores (mean 4.04, SD 0.94), reporting more pain two years after surgery, in comparison to AIS (mean 4.31, SD 0.60) (*p* = 0.043) and SK (mean 4.43, SD 0.48) patients (*p* = 0.049). Persistent postoperative pain in adolescents undergoing posterior spinal fusion is related to disease pathology while higher acute postoperative pain is associated with a more extensive surgery. Spondylolisthesis patients report more chronic pain after surgery compared to AIS and SK patients.

## 1. Introduction

Adolescent idiopathic scoliosis (AIS), Scheuermann kyphosis (SK), and spondylolisthesis are the most common indications for instrumented posterior spinal fusion (PSF) in adolescents. The primary goal of the surgical treatment for AIS and SK is to prevent progression of the deformity [1,2,3]. In spondylolisthesis, the aim is mainly to relieve pain and additionally to prevent progression or reduce sagittal deformity in high-grade slips [1]. The extension of the procedure varies between AIS, SK, and spondylolisthesis patients. Spinal deformity correction requires a long posterior instrumentation, while spondylolisthesis is treated with a single or two-level lumbar fusion [1]. Previous studies have suggested that PSF reduces pain in patients with AIS and spondylolisthesis, while pain outcomes after surgery for SK remain unclear [3,4,5,6,7].

Postoperative pain with nociceptive, neuropathic, and inflammatory components are induced from major tissue trauma after the surgical procedure [8]. Nociceptors of skin, muscles, fascia, ligaments, vertebrae, and facet joint capsules elicit the sensation of pain. Spinal surgery involves manipulation of the spine, spinal cord, and the nerve roots. These components may also suffer from inadequate perfusion and hypoxia, which may cause neuropathic pain after surgery. Tissue injury induces an inflammatory response. Damaged cells release pro-inflammatory agents such as bradykinin, histamine, cytokines, and prostanoids. Some of these inflammatory mediators act on the modulation of pain in dorsal root ganglia and directly on nociceptors. 

Persistent postoperative pain is defined as a pain on the surgical site lasting over three months after surgery, well beyond the healing process [9]. Persistent postsurgical pain is associated with increased functional disability, psychological distress, and economic costs [10]. The data from AIS patients show an incidence of postoperative persistent pain between 12 to 42% [5,11,12,13]. However, the literature regarding the persistent postsurgical pain in SK and spondylolisthesis patients is inadequate. 

The aim of our research was to determine the incidence and the predictors of acute and chronic postoperative pain in adolescents undergoing PSF. We hypothesized that acute pain is associated with the extension of surgery and that chronic postsurgical pain is more dependent on the disease pathology.

## 2. Materials and Methods

The study was approved by the Ethics Committee, Hospital District of Southwest Finland (ETMK 95/180/2011 and ETMK 38/1800/2015). Informed consent was obtained from all subjects involved in the study and from their parents if under 18 years old.

### 2.1. Study Design

The study was a retrospective analysis of a prospectively collected pediatric spine register assessing risk factors for acute and persistent pain in adolescents after instrumented PSF surgery. We reviewed 221 patients (166 AIS, 19 SK, and 36 spondylolistheses) entered consecutively into this register between March 2010 and March 2020. The date of last follow-up was 5 October 2022. This spine register is a database of children and adolescents undergoing spinal fusion surgery for AIS, SK, or spondylolisthesis at our university hospital. Eight patients were excluded from further analyses, and all of these were from the AIS group: two patients’ postoperative opioid usage was not adequately documented; two patients needed early re-operation for a neurologic deficit; one patient had a combined anteroposterior approach; one patient had chronic renal insufficiency; one patient had a concomitant neurological condition, and one patient was operated for both AIS and spondylolisthesis, leaving 213 patients for further analyses. The same skilled orthopedic spine surgeon performed surgeries on all the patients. Pain was analyzed using Scoliosis Research Society-24 (SRS-24) outcome questionnaire. The data on opioid consumption was obtained from patient records.

The anesthetic management of the AIS, SK and spondylolisthesis patients was standardized. The total intravenous anesthesia protocol has been used unchanged since 2009 in our university hospital. Anesthesia was maintained with target-controlled infusions of propofol and remifentanil titrated to maintain the bispectral index (BIS) within predetermined limits. As all patients received similar weight-based remifentanil infusions intraoperatively, this was not included in the amount of postoperative opioid reported. Muscle relaxant was not used in any of the patients at any time. Dexmedetomidine-infusion (1 µg/kg/h) was used for supplemental hypnosis and analgesia in all patients. Mean arterial pressure was maintained between 65–75 mmHg with noradrenaline infusion if needed. Normothermia was maintained. Neurophysiological measurements were done every 20 min and at certain time points. None of the patients needed postoperative ventilation. 

The majority (71%, 151/213) of the AIS, SK and spondylolisthesis patients received patient-controlled analgesia (PCA) with oxycodone for the first 48 postoperative hours. The typical oxycodone PCA contained an on-demand oxycodone-bolus of 0.03 mg/kg/dose at a maximum of three doses per hour, with no basal infusion. The patients without PCA (*n* = 62), received intravenous and oral oxycodone as needed. The patients’ oral oxycodone dosages were converted to equivalent intravenous doses (0.6 * per dose) [14]. Postoperative opioid use included all opioids received postoperatively in the postoperative anesthesia care unit (PACU), pediatric intensive care unit (PICU), and postoperative ward during the hospital stay. All patients received oral paracetamol at 15–20 mg/kg three times per day. Epidural analgesia was not used by any of the patients. A numerical rating scale (NRS) was used to evaluate pain. 

Perioperative variables collected included gender, age, height, weight, body mass index (BMI), fusion levels, surgical time, intraoperative blood loss, number of levels fused, length of hospital stay, opioid amount on first 48 h after operation, total opioid consumption during hospital stay, SRS-24 questionnaire preoperatively, six months postoperatively, and two years postoperatively.

The patients were mobilized according to our established process. The urinary catheter was removed on day 2 after surgery. The patients were requested to stand up and take a few steps on the first postoperative day. On the second postoperative day the patients were supported to walk around on the ward.

### 2.2. Scoliosis Research Society Outcome Questionnaire

The SRS-24 is a disease-specific health-related quality of life questionnaire, which is developed for scoliosis patients. The SRS-24 questionnaire is also used for other spinal surgery patients. Patients filled out the SRS-24 questionnaire preoperatively, and six and 24 months postoperatively. The SRS-24 questionnaire has seven domains: pain, general self-image, general function, general activity, postoperative self-image, postoperative function, and patient satisfaction. The scores in each field range from 1.0 to 5.0, with higher scores pointing out better patient outcomes. A score under 4 in the pain domain was considered clinically relevant and indicated moderate to severe pain [3]. The maximum score of this questionnaire is 120. The questions from 16 to 24 are related to the treatment and can therefore only be filled out after surgery. The first SRS-24 question, which asks patients to rate their pain on a scale of 1 to 9, was analyzed separately, 1 indicating no pain and 9 considered to be severe pain. A pain score over 4 was considered as moderate to severe pain. 

### 2.3. Surgical Technique

All AIS and SK patients were operated on using a posterior-only approach and had spinal cord monitoring. Bilateral segmental pedicle screw instrumentation (MESA 5.5, Stryker spine, Leesburg, VA, USA, 6.35CD Legacy or Solera 6.0, Medtronics Spinal and Biologics, Memphis, TN, USA) were used to correct the spinal deformity. Pedicle screws were inserted using a free-hand approach. Apical posterior column osteotomies were performed in all SK patients and in 46 (29.1%) of the AIS patients to facilitate deformity correction. Selection of fusion levels was according to the Lenke classification and last substantially touched vertebra for AIS [15] and stable sagittal vertebra for SK in the lumbar spine and T2 or T3 as the upper instrumented vertebra [16].

All spondylolisthesis patients had pedicle screw instrumentation with intercorporeal fusion using a TLIF cage (Crescent, Medtronic) with an autologous bone graft from the decompression. Neural elements were widely decompressed for nerve roots (L5, S1) and cauda equinae. Patients with low-grade spondylolisthesis had pedicle screws inserted into L5 and S1 to reduce the spondylolisthesis and patients with high-grade spondylolisthesis underwent instrumentation from L4 to S1 with pelvic instrumentation (iliac or S2 alar iliac screws).

### 2.4. Statistical Methods

Associations between the opioid consumption and variables (study group, gender, surgery time, intraoperative blood loss and preoperative pain) were summarized with descriptive statistics and studied one by one with the Spearman correlation (for continuous variables) and the Kruskal-Wallis test (for categorical variables). Associations between chronic pain and explanatory variables (study group, gender, surgery time, intraoperative blood loss and preoperative pain) were studied with the Spearman correlation and the Kruskal-Wallis test (three groups). Study group effects with surgical outcomes (surgery time, intraoperative blood loss and length of hospital stay) were studied with the Kruskal-Wallis test and the Dwass-Steel-Critchlow-Fligner pairwise test. Differences between study groups in the SRS-24 preoperative and two-year postoperative scores, the SRS-24 pain preoperative and two-year postoperative scores and the SRS-24 self-image and two-year postoperative scores were studied with a mixed model for repeated measures. Differences between study groups in the SRS-24 function and activity domain scores were analyzed with the Kruskal-Wallis test.

The normality of variables was assessed visually and using the Shapiro-Wilk test. In all tests, the statistical significance level was set at 0.05 (two-tailed). The analyses were carried out using the SAS system, version 9.4 for Windows (SAS Institute Inc., Cary, NC, USA).

## 3. Results

A total of 213 consecutive adolescents (146 females (69%) and 67 males (31%)) with a mean age of 15.6 years (range 10–21 years) at the time of surgery were included in this study. The cohort consisted of 158 (74.2%) AIS patients, 19 (8.9%) patients with Scheuermann kyphosis and 36 (16.9%) spondylolisthesis patients. The majority of the AIS (114 (72.2%)) and spondylolisthesis (29 (80.6%)) patients were females as opposed to the SK patients, with only 15.8% (3) females. Twenty-three (10.8%) of these patients were under the age of 13 years at the time of surgery. 

### 3.1. Surgical Outcome

There were significant differences in surgery time, intraoperative blood loss, number of levels fused, and length of hospital stay between the study groups. The Scheuermann patients had the greatest intraoperative blood loss compared to the other patient groups. The spondylolisthesis patients had the longest surgical time and the shortest hospital stay (Table 1).

### 3.2. Preoperative Pain and SRS-24 Scores

Preoperative mean (SD) SRS-24 pain scores were significantly lower in the spondylolisthesis patients, meaning more pain, 3.24 (0.9) compared to AIS 3.96 (0.7) and SK 4.02 (0.6) patients, *p* < 0.001 in both comparisons (Table 2). Seventy-six percent (25/33 patients) of the spondylolisthesis patients had a preoperative SRS pain score under 4, as compared to 42% (62/146 patients) in the AIS group and 28% (5/18 patients) in the SK group. Preoperatively 17% (25 of 146) of the AIS patients, 17% (3 of 18) of SK, and 52% (17 of 33) of the spondylolisthesis patients reported moderate to severe pain on question 1 of the SRS-24 questionnaire.

Preoperative mean (SD) SRS-24 total scores were significantly higher in AIS (4.05 (0.50)) and SK (4.02 (0.49)) patients compared to spondylolisthesis patients (3.55 (0.60)), *p* < 0.001 for both comparisons. Similarly, preoperative mean (SD) SRS-24 activity scores were significantly lower in spondylolisthesis patients, meaning less activity (3.52 (1.13)) compared to AIS (4.45 (0.81)) and SK (4.57 (0.47)) patients, (*p* < 0.001 for both comparisons).

### 3.3. Acute Postoperative Pain

There was a statistically significant difference in the opioid consumption after surgery between the three groups. The mean (SD) daily postoperative opioid consumption was significantly lower in the spondylolisthesis patients (0.36 mg/kg/day (0.17)) compared to AIS (0.51 mg/kg/day (0.25)) and SK (0.52 mg/kg/day (0.25)) patients (*p* < 0.001). Patients’ age, gender, BMI, surgery time, intraoperative blood loss or preoperative pain were not associated with increased opioid consumption postoperatively. The number of fused vertebrae correlated with the daily opioid consumption (r_s_ = 0.20, *p* = 0.0082). 

The difference in the oxycodone consumption during first 48 postoperative hours did not reach statistical significance. The mean (SD) 48 h oxycodone consumption in AIS, SK and spondylolisthesis groups were 1.68 mg/kg (1.08), 1.80 mg/kg (0.78) and 1.40 mg/kg (0.75), respectively, (*p* = 0.10). There was a correlation between longer surgical time and increased 48 h opioid consumption (r_s_ = 0.16, *p* = 0.020). Patient gender, intraoperative blood loss or preoperative pain were not associated with increased 48 h oxycodone consumption after surgery. 

### 3.4. Persistent Postoperative Pain

The SRS-24 pain domain scores showed a statistically significant improvement from preoperative levels to two-year follow-up in all three groups. This domain increased from a mean of 3.96 to 4.31 in the AIS patients (*p* < 0.001), 3.24 to 4.04 in the spondylolisthesis patients (*p* < 0.001), and 4.02 to 4.43 in the SK group (*p* = 0.03), respectively. At two years postoperatively, 11% (14 of 129) of AIS, 7% (1 of 14) of SK, and 16% (4 of 25) of the spondylolisthesis patients reported moderate to severe pain on question 1 in the SRS-24 questionnaire.

The patients with spondylolisthesis had the lowest SRS pain scores (mean 4.04, SD 0.94) at two years after surgery in comparison to AIS (mean 4.31, SD 0.60) (*p* = 0.043) and SK (mean 4.43, SD 0.48) (*p* = 0.049) patients. There was no statistical difference in SRS pain scores between the AIS and SK patients (*p* = 0.46) at the two-year follow-up. Patients’ age, gender, BMI, surgery time, intraoperative blood loss or preoperative pain were not associated with more persistent postoperative pain.

At the two-year follow-up, 27 (21%) of 129 AIS patients, 3 (20%) of 15 SK patients, and 9 (36%) of 25 patients in the spondylolisthesis group had a SRS pain score under 4 (*p* = 0.26). At the two-year follow-up there was a positive correlation between the scores in self-image and pain in the AIS patients (r_s_ = 0.30, *p* < 0.001). This correlation between self-image and pain was not seen in the other groups. 

### 3.5. Subgroup Analysis of AIS Patients

In a subgroup analysis with AIS patients only, there was a correlation between preoperative pain and daily postoperative opioid consumption (r_s_ = 0.29, *p* < 0.001) and persistent pain (*p* < 0.001). Patient gender affected the opioid consumption after surgery (*p* = 0.013) but not the persistent postoperative pain. Intraoperative blood loss, surgery time or number of fused vertebras were not associated with opioid consumption after surgery nor with persistent postoperative pain.

## 4. Discussion

To the best of our knowledge, this is among the first studies comparing persistent postoperative pain development of surgically treated AIS, SK, and spondylolisthesis patients. In our study, the extension of surgery (levels fused) predicted acute postoperative pain and greater opioid use during hospital stay. However, diagnosis and disease pathology were stronger predictors for chronic pain. 

Preoperative pain is a common risk factor for persistent pain after PSF in AIS patients [13,17,18,19]. This might explain why spondylolisthesis patients had a higher incidence of persistent pain after surgery than AIS and SK patients. Other risk factors for chronic pain after PSF in AIS patients according to the literature are child anxiety [11,18,19], longer operative time [11,20], and self-image [4]. The development of chronic postoperative pain seems to be multifactorial [8]. There is paucity in literature regarding chronic postsurgical pain after deformity correction in spondylolisthesis and SK patients. Multimodal analgesia is an important component in reducing chronic pain, but the optimal treatment protocol for adolescent spinal surgery is still not established [12,21]. Studies have shown that acute postsurgical pain predicts chronic pain in adolescents [11,22]. However, in the study conducted by Li et al. [23], it was found that opioid consumption during the acute postoperative period did not significantly predict pain six months after surgery. In our study, the mean daily postoperative opioid consumption was significantly lower in the spondylolisthesis patients compared to AIS and SK patients. Spondylolisthesis patients had more persistent pain compared to AIS and SK patients. In the subgroup analysis with AIS patients, the opioid consumption during immediate postoperative period did not predict pain six months or two years after surgery. Patients with spondylolisthesis had the lowest preoperative SRS-24 activity scores, meaning less activity compared to AIS and SK patients. They also had more preoperative pain compared to AIS and SK patients, which could lead to lower activity scores. Secondly, the conservative treatment of spondylolisthesis patients includes restriction of physical activity. Lower pain and activity scores resulted in lower preoperative SRS-24 total scores, reflecting a lower health-related quality of life in the spondylolisthesis patients compared to AIS and SK patients.

The indications for PSF differ in patients with AIS, SK, and spondylolisthesis [1,24]. Patients with AIS and SK undergo surgery for spinal deformity and spondylolisthesis patients mainly for low back and/or radicular pain. Surgical techniques also differ in nature. Multiple levels of spinal fusion are required to address deformity, while one or two-level spinal fusion with or without pelvic instrumentation is adequate to reduce or stabilize spondylolisthesis. Ideally, AIS and SK patients undergo surgery without nerve root manipulation. Wide nerve root decompression and retraction is needed in patients with spondylolisthesis. Additionally, reduction of high-grade spondylolisthesis improves possibilities of spinal union, but increases tension on L5 nerve roots [25]. Our study indicates that the extent of intraoperative tissue injury (multiple spinal fusion levels) explains relatively well immediate postoperative pain and opioid requirement after surgery. In contrast, preoperative pain and perioperative nerve root manipulation may be associated with more long-term pain, as observed in patients with spondylolisthesis.

Carreon et al. [26] determined the minimum clinically important difference (MCID) in Scoliosis Research Society-22 appearance, activity, and pain domains after surgical correction of AIS. The MCID in pain domain was 0.20. In our study, the surgical treatment of spinal deformity reduced pain after two years in all three groups of patients, and in every patient group the improvement was greater than 0.20. The greatest improvement in pain domain scores was seen in the spondylolisthesis patients. However, spondylolisthesis patients had still more chronic pain after surgery compared to AIS and SK patients.

### Limitations and Strengths

This study represents a retrospective analysis of a prospectively collected pediatric spine register with 213 consecutive adolescents with almost complete preoperative and two-year health-related quality of life data. AIS is the most common indication for spinal fusion in the adolescent age group, while the need for surgery of pediatric spondylolisthesis and Scheuermann kyphosis is much more limited. This resulted in a noticeably larger group of patients in the AIS as compared to SK and spondylolisthesis groups. Therefore, one limitation is the relatively small number of spondylolisthesis and Scheuermann kyphosis patients in the register that included 158 AIS patients but only 19 SK and 36 spondylolisthesis patients. Perioperative pain management was standardized. Health-related quality of life and pain were evaluated using the SRS-24 questionnaire, which is a validated outcome tool for adolescents undergoing instrumented posterior spinal fusion. Surgical management was standardized and included a selection of fusion levels according to the Lenke classification and last substantially touched vertebra for AIS [15], stable sagittal vertebral body for Scheuermann kyphosis [16], and one-level fusion for low-grade and two-level fusions for high-grade spondylolisthesis. All the patients were operated on by the same experienced orthopedic spine surgeon. Postoperative pain management included either oxycodone PCA or intravenous and oral oxycodone during the first 48 h postoperatively.

## 5. Conclusions

Instrumented posterior spinal fusion significantly reduced pain two years after surgery in AIS, SK, and spondylolisthesis patients. A larger number of levels fused was associated with a higher postoperative opioid consumption, as patients with AIS and SK required significantly more opioids than the patients with spondylolisthesis. In contrast, spondylolisthesis patients had more persistent pain two years after surgery compared to AIS and SK patients, suggesting that spinal pathology is more predictive of long-term pain.

## Figures and Tables

**Table 1 children-09-01729-t001:** Demographic characteristics.

Variables	AIS (*n* = 158)	SK (*n* = 19)	Spondylolisthesis (*n* = 36)	*p* Value
Sex (male:female)	44:114	3:16	7:29	<0.001
Age (years)	15.6 (2.2)	16.7 (1.3)	14.7 (1.9)	0.002
Weight (kg)	58.3 (13.5)	80.6 (27.8)	57.2 (12.5)	<0.001
BMI (kg/m^2^)	20.9 (3.9)	25.7 (7.5)	21.3 (3.7)	0.018
Surgical time (h)	2.95 (0.75)	3.45 (0.53)	3.49 (1.00)	<0.001
Intraoperative blood loos (mL)	529 (350)	616 (200)	344 (203)	<0.001
Number of levels fused (n)	10.8 (1.2)	12.9 (0.5)	2.6 (0.5)	<0.001
Daily oxycodone dose/kg during hospital stay (mg/kg/day)	0.51 (0.25)	0.52 (0.25)	0.36 (0.17)	<0.001
Major curve Preop	52 (8.5)	79 (5.6)		
Postop Length of hospital stay (day)	13 (4.6) 7.2 (1.6)	48 (8.4) 7.8 (1.6)	6.2 (2.1)	<0.001

Data presented as mean and standard deviation or number and percentage.

**Table 2 children-09-01729-t002:** SRS-24 domain scores in the groups.

SRS-24 Domains	AIS	Scheuermann Kyphosis	Spondylolisthesis	*p*-Value AIS vs. SK	*p*-Value AIS vs. Spondylolisthesis
Pain					
preoperative	3.96 (0.72)	4.02 (0.64)	3.24 (0.86)	0.82	**<0.001**
6-mth FU	4.25 (0.63)	4.36 (0.49)	3.60 (0.71)	0.46	**<0.001**
2-year FU	4.31 (0.60)	4.43 (0.48)	4.04 (0.94)	0.46	**0.04**
Self-image					
preoperative	3.83 (0.70)	3.54 (0.92)	4.00 (0.67)	0.05	0.23
6-mth FU	4.05 (0.68)	3.96 (0.50)	4.19 (0.50)	0.84	0.53
2-year FU	4.14 (0.68)	4.11 (0.61)	4.21 (0.58)	0.88	0.42
Function					
preoperative	4.03 (0.47)	3.96 (0.44)	3.84 (0.54)	0.67	0.14
6-mth FU	3.95 (0.52)	4.04 (0.69)	3.95 (0.60)	0.56	0.69
2-year FU	4.13 (0.56)	4.07 (0.42)	4.10 (0.53)	0.56	0.74
Activity					
preoperative	4.45 (0.81)	4.57 (0.47)	3.52 (1.13)	0.42	**<0.001**
6-mth FU	3.93 (0.98)	3.84 (1.10)	3.97 (1.00)	0.37	0.38
2-year FU	4.66 (0.67)	4.44 (0.88)	4.39 (1.09)	0.13	0.24
Total score					
preoperative	4.05 (0.50)	4.02 (0.49)	3.55 (0.60)	0.69	**<0.001**
6-mth FU	3.84 (0.45)	3.94 (0.46)	3.88 (0.44)	0.35	0.61
2-year FU	4.05 (0.42)	4.09 (0.46)	3.93 (0.64)	0.72	0.18

Data presented in mean (SD).

## Data Availability

The data generated and analyzed during the current study are available from the corresponding author on reasonable request.

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
