# Peer review of "The Extension of Surgery Predicts Acute Postoperative Pain, While Persistent Postoperative Pain Is Related to the Spinal Pathology in Adolescents Undergoing Posterior Spinal Fusion"

_children, 2022, doi:10.3390/children9111729_

Round 1

Reviewer 1 Report

Thank you for permitting me to review this manuscript 

In this study authors assessed predictors of acute and chronic  postoperative pain using a spine database  they found that the extension of surgery predict the acute phase and the chronic pain is related to the pathology

Table 2 please explain the p value significant differences

Here are my comments 

Methodology 

the design of the study is not ver clear for me , the authors express initially that this is a prospective study , however they speak that results are based on the review of the database , which could be interpreted as a retropective review , in addition at the end of the manuscript  , they state , that informed consent was obtained from each patient 

since these patients were adolescent and children , how informed consent obtained , was it from parents , or patients themselves , please be consistent and clarify exactly the reader on these points .

The date of the study should be reported 

there is a large difference between groups , this should be discussed

is the amount of immediate perioperative use of opioid available ? 

intraoperative + postanesthetic care unit , was this included in the assessement? 

Reviewer 2 Report

Dear Authors,

I was really pleased both to read and review this paper as one of the first studies containing this interesting subject that joins both pediatric orthopaedic surgery and anesthesiology. I was also amazed by the long-term follow-up of the patients. Introduction has clearly explained the main subject and aim of the study. Results were very well explained and shown tabulary, as well as discussed, comparing with other studies.

Keep going with your next studies,

Best regards.

Round 2

Reviewer 1 Report

The authors have improved the manuscript